# Murepavadin Enhances the Killing Efficacy of Ciprofloxacin against *Pseudomonas aeruginosa* by Inhibiting Drug Efflux

**DOI:** 10.3390/antibiotics13090810

**Published:** 2024-08-26

**Authors:** Xiaoya Wei, Dandan Zhou, Congjuan Xu, Ping Chen, Shuiping Chen, Zhihui Cheng, Yongxin Jin, Shouguang Jin, Weihui Wu

**Affiliations:** 1State Key Laboratory of Medicinal Chemical Biology, Key Laboratory of Molecular Microbiology and Technology of the Ministry of Education, Department of Microbiology, College of Life Sciences, Nankai University, Tianjin 300071, China; yyaxiaowei@mail.nankai.edu.cn (X.W.);; 2Department of Laboratory Medicine, 5th Medical Center of PLA General Hospital, Beijing 100071, China

**Keywords:** murepavadin, *Pseudomonas aeruginosa*, ciprofloxacin

## Abstract

*Pseudomonas aeruginosa* is a multidrug-resistant Gram-negative pathogen and one of the leading causes of ventilator-associated pneumonia and infections in patients with chronic obstructive pulmonary disease and cystic fibrosis. Murepavadin is a peptidomimetic that specifically targets outer-membrane lipopolysaccharide transport protein LptD of *P. aeruginosa*. In this study, we find that murepavadin enhances the bactericidal efficacy of ciprofloxacin. We further demonstrate that murepavadin increases intracellular accumulation of ciprofloxacin by suppressing drug efflux. In addition, the murepavadin–ciprofloxacin combination exhibits a synergistic bactericidal effect in an acute murine pneumonia model. In conclusion, our results identify an effective drug combination for the treatment of *P. aeruginosa* infections.

## 1. Introduction

Antimicrobial resistance (AMR) has become a major challenge in global public health. The continuous evolution of antimicrobial resistance in bacterial pathogens has caused significant global impacts on clinics [1,2]. Presently, 700,000 people die each year worldwide due to antimicrobial-resistant bacteria. It is predicted that there will be more than 10 million deaths per year attributed to antimicrobial resistance by 2050, with a global cost of around USD 100 trillion [3]. Meanwhile, the development rate of new antibiotics is declining, leaving fewer options for treating multidrug-resistant bacterial infections. To deal with this challenge, new antibiotics are needed to ensure that effective therapeutic strategies are sustained [4].

*Pseudomonas aeruginosa* is a ubiquitous Gram-negative pathogen that is responsible for high morbidity and mortality in patients with cystic fibrosis (CF) as well as community-acquired and nosocomial infections [5]. Data from China Antimicrobial Surveillance Network (CHINET) revealed that among 445,199 clinical bacterial isolates in 2023, *P. aeruginosa* was the fifth prevalent nosocomial pathogen, following *Escherichia coli*, *Klebsiella pneumoniae*, *Staphylococcus aureus*, and *Acinetobacter baumannii*. *P. aeruginosa* displays resistance to a variety of antibiotics [6]. The intrinsic resistance mechanisms include low membrane permeability, multidrug efflux pumps, chromosomally encoded β–lactamase, and biofilm formation. Mutation and acquisition of antibiotic resistance genes through horizontal transfer further increase its resistance [7,8,9]. In May 2024, the World Health Organization (WHO) updated its bacterial priority pathogens list, in which *P. aeruginosa* is listed in the high-priority group that requires development of new treatments (https://www.who.int/publications/i/item/9789240093461, accessed on 24 June 2024). 

Murepavadin (also known as POL7080) is the first peptide-based inhibitor of LptD in *P. aeruginosa* [10]. LptD is a component of the LptABCDEFG complex, which is responsible for the transport and insertion of lipopolysaccharide (LPS) into the outer membrane [1,11]. By binding to LptD, murepavadin inhibits the assembly of LPS, ultimately leading to cell death. In preclinical studies, murepavadin showed no cross-resistance with current standard-of-care antibiotics and was highly specific and potent against *P. aeruginosa* in vitro and in vivo [9,12]. In addition to its direct antibacterial activity, murepavadin may also enhance the host’s immune response by activating MRGPRX2 and induce degranulation in murine mast cells, thereby contributing to bacterial clearance and wound healing [13]. A phase III trial of intravenous murepavadin for the treatment of pneumonia was withdrawn due to severe nephrotoxicity [14]. On the contrary, aerosol administration of murepavadin displays low systemic exposure and limited toxicity, resulting in good tolerability [14,15]. Inhaled murepavadin is under a clinical trial to treat *P. aeruginosa* lung infections in cystic fibrosis patients (https://spexisbio.com/impv/, accessed on 20 June 2024). 

Ciprofloxacin is a fluoroquinolone antibiotic that is used to treat *P. aeruginosa* infections, but rising resistance frequently precludes its utilization [16,17,18]. It inhibits the activity of DNA gyrase, leading to DNA damage and cell death. In response to ciprofloxacin-induced DNA damage, RecA is activated, which triggers the cleavage of LexA and PrtR, leading to activation of the SOS response and the production of pyocins, respectively [19,20]. A novel ciprofloxacin–azithromycin sinus stent maintained a uniform coating and sustained delivery with anti-biofilm activity against *P. aeruginosa* [21]. A phase III clinical trial in non-cystic fibrosis bronchiectasis patients demonstrated that ciprofloxacin dry powder for inhalation has the potential to be an effective treatment for *P. aeruginosa* infections [22]. Another phase III clinical trial in non-cystic fibrosis bronchiectasis patients demonstrated trends towards clinical benefit with ciprofloxacin dry powder for inhalation. However, primary end-points were not met [23]. In addition, inhaled liposomal ciprofloxacin significantly delayed the median time to first pulmonary exacerbation in a phase III clinical trial in non-cystic fibrosis bronchiectasis patients with chronic *P. aeruginosa* lung infection, but not in another trial with similar settings [24]. 

Combination therapy may achieve synergistic effects and reduce the likelihood of resistance emergence and side effects [25,26]. Drug combinations may produce pharmacokinetic potentiation, where the therapeutic activity of one drug is enhanced through the influences on absorption, distribution, or metabolism by another drug [27]. For example, the combination of colistin with rifampicin was able to significantly improve the therapeutic outcomes in patients infected by carbapenemase-producing Enterobacteriaceae strains [28]. Previously, we demonstrated that murepavadin enhances the efficacy of β-lactam antibiotics against *P. aeruginosa* by disrupting outer-membrane integrity [29]. Since both inhaled ciprofloxacin and murepavadin have been evaluated in clinical trials, here, we examined the combinatorial bactericidal effects of the two antibiotics. Our results demonstrate that murepavadin enhances the cellular accumulation of ciprofloxacin and hence generates synergistic effects in vitro and in vivo. 

## 2. Results 

### 2.1. Combination of Murepavadin with Ciprofloxacin Increases the Killing Efficacy against P. aeruginosa

We first examined the bactericidal activity of murepavadin and ciprofloxacin against a reference strain, PA14. Murepavadin was used at 0.5 μg/mL, which has been shown to inhibit the growth of 99.1% of tested clinical isolates from the United States, Europe, and China [30]. The concentration of ciprofloxacin was used at 0.25 μg/mL, which is below the clinical breakpoint (0.5 μg/mL) [31]. Murepavadin and ciprofloxacin alone reduced the live bacterial numbers by approximately 50 and 500 times, respectively, eight hours after treatment. The combination further reduced the bacterial number by 1.51 × 10^4^ and 1.50 × 10^3^ times, respectively. Afterwards, bacteria treated with the single antibiotic started to grow, whereas the live bacterial number kept decreasing in the presence of the murepavadin–ciprofloxacin combination, resulting in approximately 4.50 × 10^8^- and 9.90 × 10^7^-fold fewer bacteria compared with each antibiotic alone after twenty four hours (Figure 1, Appendix A).

### 2.2. Murepavadin Enhances Intracellular Accumulation of Ciprofloxacin by Suppressing Drug Efflux

To understand the mechanism of the enhanced bactericidal effect, we examined the intracellular accumulation of ciprofloxacin. Treatment with 0.25 μg/mL murepavadin increased the intracellular level of ciprofloxacin (Figure 2a). Consistent with the ciprofloxacin accumulation results, the presence of murepavadin enhanced the ciprofloxacin-induced upregulation of *recA* and the pyocin-related genes *PA0614* and *PA0629*. Collectively, these results demonstrate that murepavadin enhances the bacterial uptake of ciprofloxacin (Figure 2b).

In *P. aeruginosa*, efflux pumps play an important role in bacterial resistance to ciprofloxacin by reducing intracellular drug concentration [17]. To examine whether murepavadin affects drug efflux, we treated bacteria with an efflux pump inhibitor, carbonyl cyanide m-chlorophenylhydrazine (CCCP) [32]. The addition of CCCP increased the intracellular ciprofloxacin amounts to the same level in the presence and absence of murepavadin (Figure 3a), indicating that murepavadin mainly increases intracellular ciprofloxacin by suppressing drug efflux. We then performed an ethidium bromide accumulation assay, which has been used to measure efflux pump activity in bacteria [33]. The presence of murepavadin increased the intracellular accumulation of ethidium bromide (Figure 3b). The addition of CCCP enhanced the intracellular ethidium bromide levels and diminished the difference in cells with or without murepavadin treatment (Figure 3b). Collectively, these results suggest that murepavadin inhibits drug efflux, which subsequently promotes the intracellular accumulation of ciprofloxacin.

### 2.3. Murepavadin Enhances Bactericidal Effects of Ciprofloxacin against P. aeruginosa Clinical Isolates

We then assessed the bactericidal effects of murepavadin in combination with ciprofloxacin against twelve clinical isolates. The strains were isolated from various infection sites and are susceptible to both of the antibiotics (Appendix A). Murepavadin or ciprofloxacin alone reduced the live bacterial number of ten strains by 14.92–1.43 × 10^4^ times eight hours after the treatment (Figure 4a–g,j–l). Afterwards, the bacteria started to grow (Figure 4a–g,j–l). Strains #26 and #123 started to grow four and six hours after the treatment of ciprofloxacin, respectively (Figure 4h,i). The combination of murepavadin and ciprofloxacin resulted in the 1.83 × 10^5^–4.03 × 10^6^-fold killing of all of the strains eight hours after the treatment, which resulted in 4.00 × 10^2^–1.78 × 10^5^-fold fewer bacteria than with each of the antibiotics alone (Figure 4). Eight hours after treatment with the combination, three out of the twelve strains started to grow (Figure 4j–l). Nevertheless, the combination reduced the bacterial number by 2.19 × 10^3^ to 4.65 × 10^8^ times compared with either antibiotic alone after twenty four hours (Figure 4). Collectively, these results demonstrate a synergistic bactericidal effect of the murepavadin–ciprofloxacin combination.

### 2.4. The Murepavadin–Ciprofloxacin Combination Displays a Synergistic Therapeutic Effect In Vivo

We next evaluated the in vivo efficacy of the murepavadin–ciprofloxacin combination against PA14 in a murine pneumonia model [34,35]. The dose of murepavadin (0.25 mg/kg) was used as previously reported in a murine pneumonia model [36]. For ciprofloxacin, the recommended daily dose for adult patients with nosocomial pneumonia is 400 mg intravenously injected every eight hours (https://reference.medscape.com/drug/cipro-xr-ciprofloxacin-342530, accessed on 18 June 2024). According to the Meeh–Rubner equation [37], the equivalent dose for intravenous injection of ciprofloxacin in mice is 52 mg/kg ciprofloxacin. In a study involving CF patients receiving a single dose of ciprofloxacin dry powder for inhalation (32.5 mg), the drug acquired a Cmax (mg/L) of 34.9 in sputum and 0.0790 in plasma [38]. In our assay, ciprofloxacin and murepavadin were administered intranasally. To minimize potential side effects, the ciprofloxacin dose was set to 0.5 mg/kg, which resulted in a similar CFU reduction to murepavadin. Treatment with murepavadin and ciprofloxacin alone decreased the bacterial loads by 33 times and 43 times, respectively, whereas the combined treatment reduced the mean bacterial load by 1171 times (Figure 5), indicating a synergistic bactericidal effect.

## 3. Discussion

ESKAPE pathogens (*Enterococcus faecium*, *S. aureus*, *K. pneumoniae*, *A. baumannii*, *P. aeruginosa*, and *Enterobacter* species) are highly antibiotic-resistant and impose a global threat to human health [39]. *P. aeruginosa* causes lung infections in patients with mechanical ventilation, cystic fibrosis, chronic obstructive pulmonary disorder (COPD), and immunodeficiency [6,40]. Nebulized antibiotics, such as tobramycin and amikacin, have been used to treat *P. aeruginosa* infections [41].

Murepavadin is a novel cyclic peptide antibiotic that specifically targets *P. aeruginosa* LptD, leading to the disruption of the asymmetry of the outer membrane [1]. LptD is a component of the LptA-G machinery that transports LPS across the periplasm and integrates it into the outer leaflet of the outer membrane. LPS is synthesized in the cytoplasm and on the inner membrane. The LptFGB complex extracts LPS from the inner membrane, followed by transportation by the LptA bridge to the outer-membrane complex, which is composed of the lipoprotein LptE and the integral β-barrel membrane protein LptD. Then, the LptD/E complex inserts LPS into the outer leaflet of the outer membrane. It has been shown that murepavadin binds to the periplasmic segment of LptD, which contains an N-terminal insert domain and a β-jellyroll domain. The binding of the antibiotic blocks LPS transport and insertion into the outer membrane [11]. In clinical trials, patients receiving inhaled murepavadin demonstrated a notable improvement in clinical symptoms and a significant reduction in bacterial burden compared with the control group [42]. The targeted delivery via inhalation facilitated direct action against the lung-colonizing bacteria, minimizing systemic toxicity. Additionally, the drug was well tolerated, with few reported adverse events. These findings underscore the potential of inhaled murepavadin as an effective and safe treatment option for patients with *P. aeruginosa* pneumonia, including those infected with drug-resistant strains. The fluoroquinolone antibiotic ciprofloxacin is commonly used to treat *P. aeruginosa* infections [17]. Clinical trials with inhaled ciprofloxacin produced promising outcomes in treating *P. aeruginosa* lung infections. Patients administered with inhaled ciprofloxacin experienced a marked reduction in bacterial load and improved clinical symptoms [43]. 

Low membrane permeability and multidrug efflux pumps play important roles in the bacterial resistance to ciprofloxacin [17]. We previously reported that murepavadin increases outer-membrane permeability, presumably owing to disruption of the LPS in the outer leaflet and alteration of the outer-membrane protein profile [29]. Here, we found that murepavadin promotes the bactericidal effect of ciprofloxacin against *P. aeruginosa.* The combination of murepavadin and ciprofloxacin synergistically reduced the bacterial load in a murine acute pneumonia model. In the ciprofloxacin uptake and ethidium bromide influx assay, the addition of CCCP diminished the effect of murepavadin, thus indicating a suppressive role of murepavadin on the multidrug efflux pumps. *P. aeruginosa* harbors at least twelve resistance-nodulation-cell division (RND) multidrug efflux pumps, which are composed of an inner-membrane protein, an outer-membrane protein, and a periplasmic protein that connects the inner- and outer-membrane proteins [44]. In *P. aeruginosa*, the pumps MexAB-OprM, MexCD-OprJ, MexEF-OprN, and MexXY-OprM play crucial roles in the resistance to fluoroquinolones [45]. Efflux through the RND pumps depends on the proton motive force [46]. We previously found that murepavadin increases the potential of the inner membrane [47]; thus, it might not affect the energy source of the pumps. It has been demonstrated that treatment with murepavadin results in the accumulation of LPS in the inner membrane [36], which might interfere with the assembly of RND pumps. Further experiments are needed to examine the expression and assembly of the pumps. In addition, increased outer-membrane permeability might also contribute to the murepavadin-promoted bactericidal effect of ciprofloxacin. Treatment with murepavadin might increase membrane permeability while reducing the efflux systems, which may sensitize bacteria to ciprofloxacin or other antibiotics. Thus, sequential treatment with murepavadin and ciprofloxacin might further increase bactericidal efficacy. Further experiments are warranted to evaluate the therapeutic effect.

In this study, we used the PA14 strain to elucidate the mechanism of synergy and evaluate in vivo efficacy. PA14 is a burn wound isolate and has attracted much attention due to its high toxicity in plants and animals [48]. It has been wildly used as a *P. aeruginosa* reference strain [49]. Since its genome sequence is available, PA14 has been used in numerous mechanistic studies and animal models. The clinical strains were isolated from various infection sites. The MICs of ciprofloxacin for the strains are comparable to wild type PA14 (within a 2-fold difference). Thus, the uptake and efflux of ciprofloxacin of those strains might be similar to wild type PA14. Clinical isolates are different in genotypes, leading to variations in bacterial pathogenesis and responses to antibiotics. However, the genotypes of the stains were not examined. Since the therapeutic effect is also affected by bacterial virulence, biofilm formation, etc., the phenotypes and the in vivo treatment efficacy warrant further evaluation in the future to verify the effectivity of the combination.

We investigated the synergistic effects of the murepavadin–ciprofloxacin combination against ciprofloxacin-susceptible strains. Additionally, we tested strains that are resistant to ciprofloxacin (with MIC higher than the break point, 2 μg/mL). No synergistic effect was observed with the combination. In clinical isolates, one of the major resistance mechanisms is mutation in the DNA gyrase gene *gyrA*, which reduces the drug binding affinity. Additionally, the upregulation of efflux systems contributes to resistance [44,50]. Although murepavadin interferes with the efflux systems, the increased accumulation of intracellular ciprofloxacin might not be able to overcome the reduced affinity, rendering no synergistic effect. Therefore, it is likely that the ciprofloxacin-resistant strain we tested harbors mutation in the *gyrA* gene. Further experiments are needed to elucidate the resistance mechanism of the strain. Other antibiotics can be used in combination with murepavadin to test treatment efficacy.

The combination of β-lactam and β-lactase inhibitors is a common combination with antipseudomonal activity, such as ceftolozane–tazobactam and ceftazidime–avibactam [51,52]. Chronic infection of *P. aeruginosa* in cystic fibrosis patients is usually treated with nebulized tobramycin or aztreonam, which improves lung function and reduces acute exacerbation of CF [53]. Compared with this clinical practice, murepavadin exhibits high effectivity against a wide range of *P. aeruginosa* clinical isolates, including carbapenem-resistant strains [1]. A phase I study of inhaled murepavadin has been initiated (https://spexisbio.com/impv/, accessed on 24 May 2024), and this is of great significance for managing chronic lung infections, especially *P. aeruginosa* infections in CF patients. Clinical trials have shown that inhaled ciprofloxacin is effective in treating chronic respiratory infections caused by *P. aeruginosa* [22,23]. Due to the potential formation of persistent bacteria in chronic infections that are difficult to eradicate, there is an urgent need for inhaled antibiotic strategies to achieve high airway concentrations and minimal systemic toxicity, in order to prevent further lung damage [42]. At present, the options of inhaled antibiotics on the market are limited, and the successful development and application of murepavadin and ciprofloxacin may provide new treatment options for such patients. 

A phase III clinical trial of intravenous injection of murepavadin was halted due to its nephrotoxicity, and nebulized antibiotics are still in clinical trials [14]. It has been reported that ciprofloxacin caused acute renal failure, demonstrating potential nephrotoxicity [54,55]. Therefore, the safety of the murepavadin–ciprofloxacin combination needs to be carefully evaluated. 

Overall, our results suggest that the combination of nebulized murepavadin and ciprofloxacin might be beneficial for the treatment of human lung infections caused by *P. aeruginosa*.

## 4. Materials and Methods

### 4.1. Bacterial Strains and Growth Conditions

The bacterial strains and primers utilized in this study are shown in Appendix A. *P. aeruginosa* strains were cultivated in Mueller–Hinton broth (MHB) or Lysogeny broth (LB) medium at 37 °C with under 200 rpm shaking.

### 4.2. Minimal Inhibitory Concentration (MIC) Determination

Bacteria were cultured in LB to an OD_600_ of 1.0. The Clinical and Laboratory Standards Institute (CLSI) guidelines were followed to measure the minimal inhibitory concentrations (MICs) of the indicated antibiotics in Cation-adjusted Mueller–Hinton Broth (CA-MHB, QDRS Biotec, Qingdao, China) [31]. The MICs were examined in triplicate. Murepavadin trifluoroacetic acid and ciprofloxacin were purchased from MCE (MedChemExpress, Shanghai, China) and Sangon Biotech (Shanghai, China), respectively.

### 4.3. Time-Dependent Killing Assays

Bacteria were cultured to an OD_600_ of 1.0 in LB. After being diluted ten-fold into 2 mL of fresh MHB, the bacteria were treated with the specified antibiotics at 37 °C with shaking. At indicated intervals (0, 2, 4, 6, 8, and 24 h), viable bacterial counts were determined by plating on LB agar. All experiments were conducted in triplicate.

### 4.4. Intracellular Accumulation of Ciprofloxacin

The uptake of ciprofloxacin was measured as described previously, with slight modifications [56]. Briefly, PA14 was cultured in MHB at 37 °C with shaking until the OD_600_ reached 1.0. For each individual bacterial aliquot (2.5 mL), a final concentration of 10 μg/mL ciprofloxacin was added with or without 0.25 µg/mL murepavadin and 40 µg/mL CCCP, followed by incubation with shaking for 15 min. Then, 300 µL of each sample was collected for OD_600_ measurement. The remaining bacteria were collected by centrifugation at 3000× *g* at 4 °C for 7 min and washed three times with cold PBS to remove extracellular ciprofloxacin. The bacterial cells were then resuspended in 0.1 M glycine–HCl buffer (pH 3.0). The samples were kept overnight in the dark to enable bacterial lysis and ciprofloxacin release. After centrifugation at 20,000× *g* for 7 min, the fluorescence of ciprofloxacin in the supernatant was examined by utilizing a microplate reader (BioTek Synergy H1, Agilent, Santa Clara, CA, USA) with excitation at 275 nm and emission at 450 nm. The intracellular ciprofloxacin levels were normalized to OD_600_.

### 4.5. Ethidium Bromide Influx Assay

The ethidium bromide influx assay was carried out as described previously, with slight modifications [57,58]. Bacteria were cultured in LB to an OD_600_ of 0.5. A total of 1 mL of the bacterial culture was collected with centrifugation and resuspended in an equal volume of PBS. The bacteria were incubated at 37 °C for 30 min with 0.25 μg/mL murepavadin and 40 µg/mL CCCP alone or in combination, in the presence of 2 μg/mL ethidium bromide. Subsequently, the bacteria were rinsed twice and resuspended in 0.2 mL of PBS. The fluorescence of ethidium bromide (excitation wavelength of 530 nm and emission wavelength of 585 nm) was detected with a microplate reader (Biotek Synergy H1, Agilent, Santa Clara, CA, USA). The relative uptake of ethidium bromide was normalized to the OD_600_ of each sample.

### 4.6. RNA Isolation and Quantitative Real-Time PCR (qRT-PCR)

The overnight bacterial cultures were diluted 100-fold into MHB and incubated with murepavadin (0.03125 μg/mL) and ciprofloxacin (0.025 μg/mL), either alone or in combination, at 37 °C until the OD_600_ reached 1.0. The bacteria were harvested by centrifugation. The total bacterial RNA was extracted with a Bacteria Total RNA Kit (Zoman Biotec, Beijing, China). cDNA was synthesized from 1 μg of RNA with PrimeScript Reverse Transcriptase (TransGen biotech, Beijing, China) in a volume of 20 μL. Afterwards, the cDNA was mixed with SYBR Premix (Vazyme, Nanjing, China) and specific primers (Appendix A). The *rpsL* gene that encodes the 30S ribosomal protein was used as an internal reference. 

### 4.7. Murine Lung Infection Model

The mouse acute pneumonia model was conducted as previously described [34,35]. PA14 was cultivated to an OD_600_ of 1.0 in LB. The bacteria were washed twice with 0.9% NaCl and resuspended to a concentration of 2 × 10^8^ CFU/mL. Four groups of eight female BALB/c mice (aged 6–8 weeks) (Vital River) were anesthetized via intraperitoneal injection with 90 μL of 7.5% chloral hydrate. A total of 20 μL of the bacterial suspension was inoculated intranasally to each mouse. Three hours post-infection (hpi), the mice were anesthetized again and treated intranasally with either 20 μL of NaCl alone or NaCl containing murepavadin (0.25 mg/kg), ciprofloxacin (0.5 mg/kg), or a combination of both antibiotics at the concentrations used for monotreatment. After 13 h, the mice were sacrificed by CO_2_ inhalation. The lungs of the mice were homogenized in sterile 1% proteose peptone (Solarbio, Beijing, China). The bacterial burdens were quantified through serial dilution and plating.

## Figures and Tables

**Figure 1 antibiotics-13-00810-f001:**
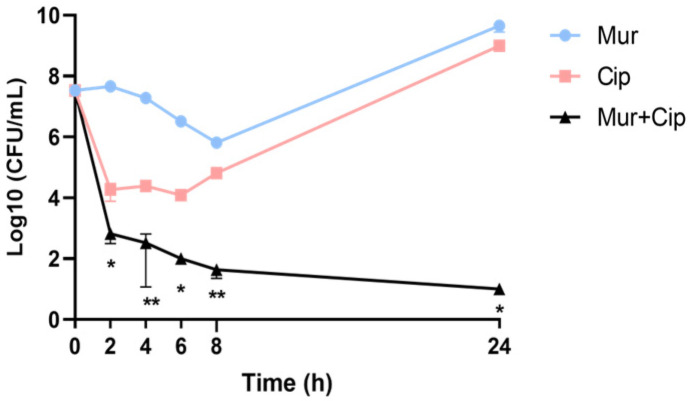
The time–killing curves of murepavadin (0.5 μg/mL) and ciprofloxacin (0.25 μg/mL) alone and in combination against PA14. *, *p* < 0.05; **, *p* < 0.01 compared with the bacteria treated with the single antibiotic at the corresponding time points by Student’s *t*-test. The data are the means ± SD of three independent samples. Mur, murepavadin; Cip, ciprofloxacin.

**Figure 2 antibiotics-13-00810-f002:**
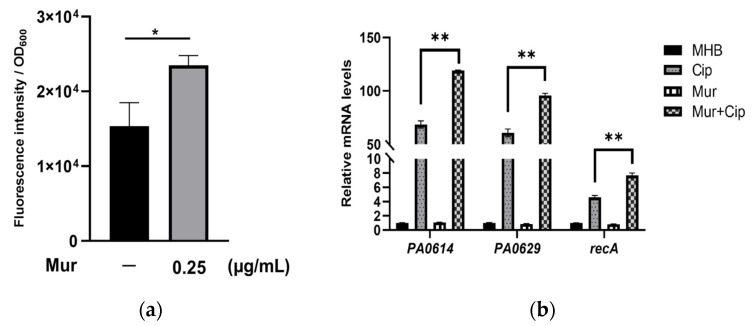
Murepavadin increases the intracellular accumulation of ciprofloxacin. (**a**) Wild type PA14 was treated with 10 µg/mL ciprofloxacin with or without 0.25 µg/mL murepavadin in Mueller–Hinton Broth (MHB) for 15 min, followed by measurement of fluorescence of intracellular ciprofloxacin. The relative fluorescence was normalized by the OD_600_. (**b**) PA14 cells were treated with or without murepavadin (0.03125 μg/mL) and ciprofloxacin (0.025 μg/mL), individually or in combination and grown to OD_600_ = 1 at 37 °C. The mRNA levels of *PA0614*, *PA0629*, and *recA* were determined by qRT-PCR. The data are the means ± SD of three independent samples. *, *p* < 0.05; **, *p* < 0.01 by Student’s *t*-test. Mur, murepavadin; Cip, ciprofloxacin.

**Figure 3 antibiotics-13-00810-f003:**
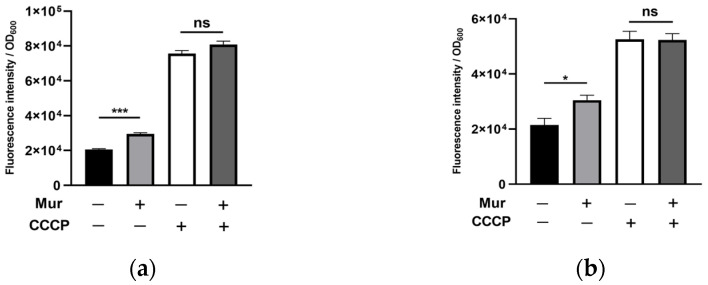
Effects of CCCP on intracellular accumulation of ciprofloxacin and ethidium bromide. (**a**) Wild type PA14 was treated with 10 µg/mL ciprofloxacin with or without 0.25 µg/mL murepavadin and 40 µg/mL CCCP in MHB for 15 min, followed by measurement of the fluorescence of intracellular ciprofloxacin. (**b**) Wild type PA14 was incubated with or without 0.25 µg/mL murepavadin, 40 µg/mL CCCP and 2 μg/mL ethidium bromide in PBS for 30 min at 37 °C. The relative fluorescence was normalized by the OD_600_. ns, not significant; *, *p* < 0.05; ***, *p* < 0.001 by Student’s *t*-test. Mur, murepavadin.

**Figure 4 antibiotics-13-00810-f004:**
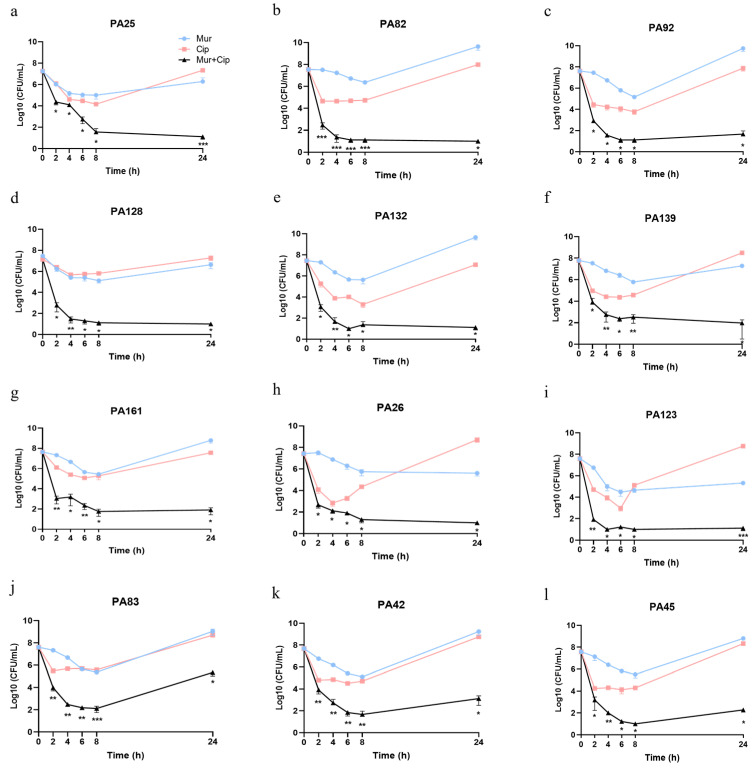
Bactericidal effects of murepavadin in combination with ciprofloxacin against clinical strains. (**a**–**l**) The bacteria were treated with murepavadin (0.5 μg/mL) and ciprofloxacin (0.25 μg/mL) individually or in combination at 37 °C with shaking. At the indicated time points, the live bacteria numbers were determined by plating. The strain numbers were shown in each of the figures. *, *p* < 0.05; **, *p* < 0.01; ***, *p* < 0.001 compared with the bacteria treated with the single antibiotic at the corresponding time points by Student’s *t*-test. The data are the means ± SD of three independent samples. Mur, murepavadin; Cip, ciprofloxacin.

**Figure 5 antibiotics-13-00810-f005:**
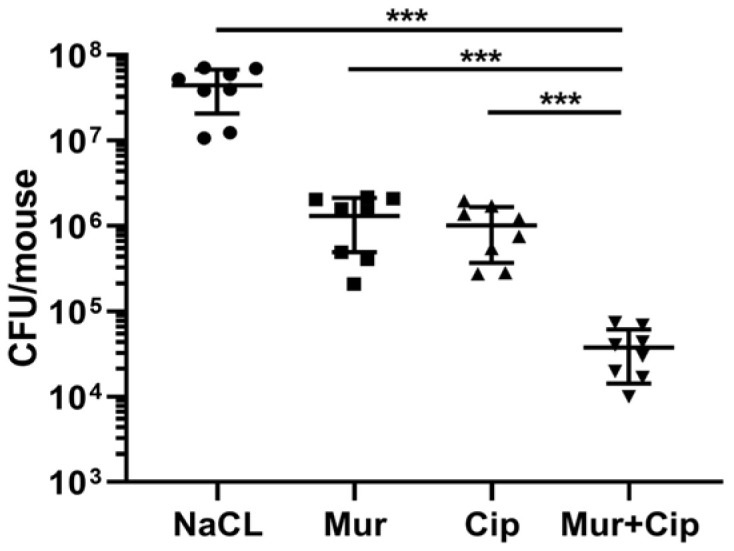
The murepavadin–ciprofloxacin combination displayed synergistic effect in vivo. Mice were inoculated intranasally with 4 × 10^6^ CFU of wild type PA14. Three hours later, NaCl, murepavadin, ciprofloxacin, or the murepavadin–ciprofloxacin combination were administered intranasally. Sixteen hours post-infection, the bacterial loads in the lungs were determined. ***, *p* < 0.001 by Student’s *t*-test. Mur, murepavadin; Cip, ciprofloxacin.

## Data Availability

The data that support the findings of this study are available from the corresponding author upon reasonable request.

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
