# Peer review of "Murepavadin Enhances the Killing Efficacy of Ciprofloxacin against *Pseudomonas aeruginosa* by Inhibiting Drug Efflux"

_antibiotics, 2024, doi:10.3390/antibiotics13090810_

Round 1
Reviewer 1 Report
Comments and Suggestions for Authors
The authors combine murepavadin with ciprofloxacin as a synergic therapy against P. aeruginosa. The comments regarding this manuscript are enlisted as follows:
- Figure 4 presents low quality and is complicated visualize it. Improve the resolution or adjust the size of this.
- The introduction need to be completed in terms of antibiotic studies, for example with some of these manuscripts: https://doi.org/10.3390/antibiotics12020279; https://doi.org/10.3390/nanomanufacturing2030010
- The manuscript needs a molecular explanation regarding the mechanism of murepavadin uptake, in other words, expand the discussion of this section.
Author Response
Comment 1: Figure 4 presents low quality and is complicated visualize it. Improve the resolution or adjust the size of this.
Response 1:Thank you for pointing out this problem. We have increased image’s size for better visualization.
Comment 2: The introduction need to be completed in terms of antibiotic studies, for example with some of these manuscripts: https://doi.org/10.3390/antibiotics12020279IF: 4.3 Q1 ; https://doi.org/10.3390/nanomanufacturing2030010
Response 2: Thank you for your advice. We added the following contents to the introduction section: The continuous evolution of antimicrobial resistance in bacterial pathogens has caused significant global impacts on clinics (reference #1, 2 in the manuscript). Meanwhile, the development rate of new antibiotics is declining, leaving fewer options for treating multidrug-resistant bacterial infections. To deal with this challenge, new antibiotics are needed to ensure that effective therapeutic strategies are sustained (reference #4 in the manuscript).(lines 23-25, 28-30)
In preclinical studies, murepavadin showed no cross-resistance with current standard-of-care antibiotics and was highly specific and potent against P. aeruginosa in vitro and in vivo (reference #9, 12 in the manuscript). In addition to its direct antibacterial activity, murepavadin may also enhance the host's immune response by activating MRGPRX2 and induces degranulation in murine mast cells, thereby contributing to bacterial clearance and wound healing (reference #13 in the manuscript). A phase III trial of intravenous murepavadin for the treatment of pneumonia was withdrawn due to the severe nephrotoxicity (reference #14 in the manuscript). (lines 47-53)
Ciprofloxacin is a fluoroquinolone antibiotic that is used to treat P. aeruginosa infections, but rising resistance frequently precludes its utilization (reference #16-18 in the manuscript). A novel ciprofloxacin-azithromycin sinus stent maintained a uniform coating and sustained delivery with anti-biofilm activities against P. aeruginosa (reference #21 in the manuscript). (lines 57-58, 61-63,)
Drug combinations may produce pharmacokinetic potentiation, where the therapeutic activity of one drug is enhanced through the influences on absorption, distribution or metabolism by another drug (reference #27 in the manuscript). (lines 72-74)
Comment 3:The manuscript needs a molecular explanation regarding the mechanism of murepavadin uptake, in other words, expand the discussion of this section.
Response 3: Thank you for your suggestion. We added the following description in the Discussion section. LptD is a component of the LptA-G machinery that transports LPS across the periplasm and integrates it into the outer leaflet of the outer membrane. LPS is synthesized in the cytoplasm and on the inner membrane. LptFGB complex extracts LPS from the inner membrane, followed by transportation by the LptA bridge to the outer membrane complex that is composed of the lipo-protein LptE and the integral β-barrel membrane protein LptD. Then the LptD/E complex inserts LPS into the outer leaflet of the outer membrane. It has been shown that murepavadin binds to the periplasmic segment of LptD that contains an N-terminal insert domain and a β-jellyroll domain. The binding of murepavadin blocks LPS transport and insertion into the outer membrane (reference #11 in the manuscript). (lines 190-198)
Reviewer 2 Report
Comments and Suggestions for Authors
The article by Wei et al. entitled “Murepavadin enhances the killing efficacy of ciprofloxacin against Pseudomonas aeruginosa by inhibiting drug efflux” shows the drug combination's effect on inhibiting bacterial growth. This is an interesting study where authors demonstrated that a murepavadin-ciprofloxacin combination effectively reduces bacterial growth. The paper is well-written and easy to follow up. The experimental setup is simple, and the results presented are able to support the hypothesis/question.
I have the following comments/ concerns that need to be addressed.
1. Figure 1; Please clarify that the effects of individual drugs that are shown in Figure 1 are bactericidal vs bacteriostatic. In the present study, the authors administered the treatments simultaneously. Did they consider using Murepavadin alone for a few hours before introducing the second drug? This could improve the Ciprofloxacin effect even further.
2. Please include data for conc. titration for both antibiotics to show that the concentration used is not lethal to the cell. The actual graph of MIC would be helpful and it can be included in supplementary.
3. Figure 2; Please mention the full name if they are being used for the first time. What is MHB?
4. Figure 4: WT control is missing in the assay showing the effect of the combined drug on clinical isolates. Please includes.
Author Response
Comment 1: Figure 1; Please clarify that the effects of individual drugs that are shown in Figure 1 are bactericidal vs bacteriostatic. In the present study, the authors administered the treatments simultaneously. Did they consider using Murepavadin alone for a few hours before introducing the second drug? This could improve the Ciprofloxacin effect even further.
Response 1: In figure 1, the concentration of individual antibiotic has bactericidal effect. We revised the description for clarification. As you suggested, it will be interesting to compare the difference between combination therapy and sequential therapy. We added discussion in the Discussion section. (lines 89-91, 228-231)
Comment 2:Please include data for conc. titration for both antibiotics to show that the concentration used is not lethal to the cell. The actual graph of MIC would be helpful and it can be included in supplementary.
Response 2: The concentration of ciprofloxacin in the influx assay was used as previously reported (reference #56 in the manuscript). After the treatment, the extracellular ciprofloxacin was removed by washing with PBS. Fluorescence of the intracellular ciprofloxacin was normalized to bacterial OD600. Therefore, this assay measures the intracellular level of ciprofloxacin in intact cells (Fig. 2A), although the cells might not be able to form colonies on plates. In the experiment of Fig. 2B, the overnight bacterial cultures were diluted 100-fold in fresh medium with or without the antibiotics at the concentrations below the MICs. The bacteria were able to grow to OD600 of 1.0. Therefore, the isolated RNA are from live bacterial cells.
The MICs of murepavadin and ciprofloxacin were shown in the supplementary Table S2.
Comment 3:Figure 2; Please mention the full name if they are being used for the first time. What is MHB?
Response 3:Thank you for pointing this out. We have added the full name of MHB medium in lines 112-113.
Comment 4:Figure 4: WT control is missing in the assay showing the effect of the combined drug on clinical isolates. Please includes.
Response 4:PA14 is a P. aeruginosa wild type reference strain and we detected the effect of the combined antibiotics against PA14 in figure 1 (reference #49 in the manuscript).
Reviewer 3 Report
Comments and Suggestions for Authors
The manuscript by Weihui Wu et al. investigates the synergistic effects of murepavadin and ciprofloxacin against Pseudomonas aeruginosa. The study provides evidence that murepavadin enhances the bactericidal efficacy of ciprofloxacin by inhibiting drug efflux, leading to increased intracellular accumulation of ciprofloxacin. The paper also explores the synergistic effects of this combination in both in vitro and in vivo settings. The topic is significant given the increasing prevalence of antibiotic resistance in P. aeruginosa, and the findings could contribute to the development of more effective treatment strategies. However, the authors need to address the following key points.
· Please address how the combination of murepavadin with ciprofloxacin compares to other known combinations or treatments currently in clinical practice. Try to elaborate on the potential clinical implications, including any advantages over existing therapies.
· The selection criteria for the clinical isolates are not fully described, making it difficult to assess the representativeness of the isolates used in the study. Additionally, while the MIC values for both antibiotics are reported, the variability across different strains is not adequately discussed.
· While the clinical isolates are included, the data presented primarily focuses on single strain, PA14, limiting the generalizability of the findings. The authors need to provide a better explanation.
· The authors need to mention the potential limitations of the study as well as the alternative explanations for the observed synergistic effects.
· The manuscript would benefit from a discussion on the potential toxicity or side effects of the combination therapy, which is not currently addressed.
Addressing these key points will enhance the manuscript's impact and contribution to the field of antimicrobial resistance research.
Author Response
Comment 1:Please address how the combination of murepavadin with ciprofloxacin compares to other known combinations or treatments currently in clinical practice. Try to elaborate on the potential clinical implications, including any advantages over existing therapies.
Response 1:Combination of β-lactam and β-lactase inhibitors is a common combination with antipseudomonal activity, such as ceftolozane-tazobactam and ceftazidime-avibactam (reference #51, 52 in the manuscript). Chronic infection of P. aeruginosa in cystic fibrosis patients is usually treated with nebulized tobramycin or aztreonam, which improves lung function and reduces acute exacerbation of CF (reference #53 in the manuscript). Compared to these clinical practice, murepavadin exhibits high effectivities to a wild range of P. aeruginosa clinical isolates, including carbapenem resistant strains (reference #1 in the manuscript). A Phase I study of inhaled murepavadin has been initiated (https://spexisbio.com/impv/) and this is of great significance for managing chronic lung infections, especially P. aeruginosa infections in CF patients. Clinical trials have shown that inhaled ciprofloxacin is effective in treating chronic respiratory infections caused by P. aeruginosa (reference #22, 23 in the manuscript). Due to the potential formation of persistent bacteria in chronic infections that were difficult to eradicate, there is an urgent need for inhaled antibiotic strategies to achieve high airway concentrations and minimal systemic toxicity, in order to prevent further lung damage (reference #42 in the manuscript). At present, the options of inhaled antibiotics in the market are limited, and the successful development and application of murepavadin and ciprofloxacin may provide new treatment options for such patients. We added the content to the discussion section. (lines 255-269)
Comment 2:The selection criteria for the clinical isolates are not fully described, making it difficult to assess the representativeness of the isolates used in the study. Additionally, while the MIC values for both antibiotics are reported, the variability across different strains is not adequately discussed.
Response 2:Thank you for pointing this out. We chose the strains that are susceptible to both murepavadin with ciprofloxacin. The strains were isolated from various infection sites. The MICs for the strains are comparable to the wild type PA14 (within 2-fold difference). Thus the uptake and efflux of ciprofloxacin of those strains might be similar to wild type PA14. However, the genotypes of the stains were not examined. Since the therapeutic effects is also affected by bacterial virulence and biofilm formation, etc, the phenotypes and the in vivo treatment efficacies can be further evaluated in the future. We added the discussion to the Discussion section. (lines 142-143, 236-239, 240-243)
Comment 3:While the clinical isolates are included, the data presented primarily focuses on single strain, PA14, limiting the generalizability of the findings. The authors need to provide a better explanation.
Response 3:PA14 (UCBPP-PA14) is a burn wound isolate and has attracted much attention due to its high toxicity in plants and animals (reference #48 in the manuscript). It has been wildly used as a P. aeruginosa reference strain (reference #49 in the manuscript). Since its genome sequence is available, PA14 has been used in numerous mechanistic studies and animal models. In this study, we used PA14 to elucidated the mechanism of synergy and evaluate the in vivo efficacy. Clinical isolates are different in genotypes, leading to variations in bacterial pathogenesis and responses to antibiotics. Further tests with clinical isolates in the pneumonia model are needed to verify the effectivity of the combination. We added the discussion to the Discussion section. (lines 232-236,239-243)
Comment 4:The authors need to mention the potential limitations of the study as well as the alternative explanations for the observed synergistic effects.
Response 4:Thank you for the suggestion. In our assay, we investigated the in vitro and in vivo therapeutic effects of the murepavadin-ciprofloxacin combination against P. aeruginosa strains. The strains used are susceptible to ciprofloxacin. We also tested strains that are resistant to ciprofloxacin (with MIC higher than the break point, 2 μg/mL). No synergistic effect was observed with the combination (data not shown). In clinical isolates, one of the major resistance mechanisms is mutation in the DNA gyrase gene gyrA, which reduces the drug binding affinity (reference #50 in the manuscript). Additionally, upregulation of efflux systems contributes to the resistance (reference #44 in the manuscript). Although murepavadin interferes with the efflux systems, the increased accumulation of intracellular ciprofloxacin might not be able to overcome the reduced affinity, rendering no synergistic effect. Therefore, it is likely that the ciprofloxacin resistant strain we tested harbors mutation in the gyrA gene. Further experiments are needed to elucidate the resistance mechanism of the strain. Other antibiotics can be used in combination with murepavadin to test treatment efficacies. We added the discussion to the Discussion section. (lines 244-254)
Comment 5:The manuscript would benefit from a discussion on the potential toxicity or side effects of the combination therapy, which is not currently addressed.
Response 5:Thank you for the suggestion. A phase III clinical trial of intravenous injection of murepavadin was halted due to its nephrotoxicity, and nebulized antibiotics are still in clinical trials (reference #14 in the manuscript). It has been reported that ciprofloxacin caused acute renal failure, demonstrating a potential nephrotoxicity (reference #54, 55 in the manuscript). Therefore, the safety of the murepavadin-ciprofloxacin combination needs to be carefully evaluated. We added the discussion to the Discussion section. (lines 270-274)
Reviewer 4 Report
Comments and Suggestions for Authors
This study addresses the critical need for novel therapeutic approaches for multidrug-resistant Pseudomonas aeruginosa. In this vein, the authors explored the effect of murepavadin, that is peptidomimetic specifically targeting the LptD protein of P. aeruginosa. Murepavadin intensified the bactericidal efficacy of ciprofloxacin by increasing its intracellular accumulation and suppressing drug efflux. It seems to be is a synergetic effect strongly supported by experimental evidence.The authors develop a preclinical murine pneumonia model for validating successfully this synergetic effect.
Yett, murepavadin enhances bactericidal effects of ciprofloxacin against P. aeruginosa in 12 studied clinical isolates.
The murepavadin-ciprofloxacin combination displays also a synergistic therapeutic effect in vivo, involving CF patients receiving a single dose of ciprofloxacin 153 dry powder for inhalation.
The study’s potential should influence future research and novel therapeutic approaches for multidrug-resistant Pseudomonas aeruginosa. The article conclusions are well-supported based and analyzing bibliographical data.
The tables and figures in the article are well conceived and well prepared and contribute to the paper’s clarity and impact.
My suggestion is to ACCEPT this opinion article for publication.
Author Response
Response : We are very grateful to your professional reviewwork on our article.